# Progress in the Management of Paediatric-Onset Multiple Sclerosis

**DOI:** 10.3390/children7110222

**Published:** 2020-11-09

**Authors:** Aphra Luchesa Smith, Christina Benetou, Hayley Bullock, Adam Kuczynski, Sarah Rudebeck, Katie Hanson, Sarah Crichton, Kshitij Mankad, Ata Siddiqui, Susan Byrne, Ming Lim, Cheryl Hemingway

**Affiliations:** 1University College London Medical School, London WC1E 6DE, UK; zchaabl@ucl.ac.uk; 2Children’s Neurosciences, Evelina London Children’s Hospital, Guy’s and St Thomas’ NHS Foundation Trust, London SE1 7EH, UK; christina.benetou@nhs.net (C.B.); sarah.rudebeck@gstt.nhs.uk (S.R.); sarah.crichton@gstt.nhs.uk (S.C.); ata.siddiqui@gstt.nhs.uk (A.S.); susan.byrne@gstt.nhs.uk (S.B.); 3Department of Neurology, Great Ormond Street Hospital for Children, London WC1N 3JH, UK; hayley.bullock@gosh.nhs.uk (H.B.); katie.hanson@gosh.nhs.uk (K.H.); 4Department of Neuropsychology, Great Ormond Street Hospital for Children, London WC1N 3JH, UK; adam.kuczynski@gosh.nhs.uk; 5Department of Radiology, Great Ormond Street Hospital for Children, London WC1N 3JH, UK; kshitij.mankad@gosh.nhs.uk; 6Department of Neuroradiology, King’s College Hospital, London SE5 9RS, UK; 7Department of Women and Children’s Health, School of Life Course Sciences, Faculty of Life Sciences and Medicine, King’s College London, London SE5 9NU, UK

**Keywords:** demyelination, disease-modifying treatment, neurocognitive, neurodisability, relapse

## Abstract

Considerable progress has been made in the understanding and treatment of paediatric-onset multiple sclerosis (POMS); how this has translated into more effective care is less well understood. Here, we evaluate how recent advances have affected patient management and outcomes with a retrospective review of POMS patients managed at two paediatric neuroimmunology centres. Two cohorts, seen within a decade, were compared to investigate associations between management approaches and outcomes. Demographic, clinical and neurocognitive data were extracted from case notes and analysed. Of 51 patients, 24 were seen during the period 2007–2010 and 27 during the period 2015–2016. Median age at onset was 13.7 years; time from symptom onset to diagnosis was 9 months. Disease-modifying therapies were commenced in 19 earlier-cohort and 24 later-cohort patients. Median time from diagnosis to treatment was 9 months for earlier vs. 3.5 months in later patients (*p =* 0.013). A wider variety of treatments were used in the later cohort (four medications earlier vs. seven in the later and two clinical trials), with increased quality of life and neurocognitive monitoring (8% vs. 48% completed PedsQL quality of life inventory; 58% vs. 89% completed neurocognitive assessment). In both cohorts, patients were responsive to disease-modifying therapy (mean annualised relapse rate pre-treatment 2.7 vs. 1.7, mean post-treatment 0.74 vs. 0.37 in earlier vs. later cohorts). In conclusion, over the years, POMS patients were treated sooner with a wider variety of medications and monitored more comprehensively. However, this hugely uncontrolled cohort did not allow us to identify key determinants for the improvements observed.

## 1. Introduction

Multiple sclerosis (MS) is a chronic, progressive auto-immune neurological disorder characterised by a relapsing–remitting course [1]. Chronic inflammation leads to neuronal degeneration, axonal loss and disability accumulation. The precise cause of MS is not fully understood; however, many possible contributing factors have been identified, including genetics, Epstein–Barr virus (EBV) infection and vitamin D deficiency [2]. Unlike adult-onset MS, paediatric-onset MS (POMS) is rare, with an estimated incidence of 0.99 per 100,000 in under 18s [3].

Paediatric presentations are similar to adults, with two-thirds of paediatric patients presenting with multiple symptoms [4]. Children have a more active disease course than adults, with both a higher relapse rate and greater lesion load on neuroimaging [5,6]. However, presumably due to enhanced neuroplasticity, physical disability progresses more slowly than in adults [7], with cognitive impairment the predominant form of disability [8,9]. Despite neuroplasticity, children still reach similar levels of disability (wheel-chair dependence) at a younger age than their adult counterparts, due to their earlier age of onset [10]. Quality of life is often significantly reduced with implications for school, social and physical functioning and greater comorbid mental health difficulties for both children and their families [11,12].

Early treatment is recognised to improve adult long-term outcomes [13], and previous challenges in managing children with MS included significant delays in diagnosis and access to treatments [14]. Advances in recognition and management of POMS resulted from the International Paediatric Multiple Sclerosis Study Group’s (IPMSSG) efforts in developing a diagnostic and management consensus, which strongly promoted centres of excellence staffed by paediatric specialists [15,16]. Our aims were to characterise how POMS treatment has changed from 2007 to 2017, and how this has impacted on disease burden and cognitive outcomes.

## 2. Materials and Methods

### 2.1. Participants

This study was based at Great Ormond Street Hospital (GOSH) and Evelina London Children’s Hospital (ELCH), where multidisciplinary demyelinating disease clinics were first established in July 2007. All patients were under 18, with a recorded diagnosis of relapsing-remitting MS. Patients were re-evaluated carefully to exclude MS mimics, such as myelin oligodendrocyte glycoprotein antibody associated disease (MOGAD) and AQP4-Ab Neuromyelitis Optica Spectrum Disorder, that are phenotypically distinct from MS [17]. To map changing management and clinical outcome of patients, two cohorts at the extremes of the time period were identified, with the time epoch adjusted to obtain comparable numbers in each group. One cohort consisted of all patients first seen in the period 2007–2010 (from here on referred to as the earlier cohort); the other, all patients initially seen in the period 2015–2016 (referred to as the later cohort).

### 2.2. Procedure

Clinical data already collected as part of standard clinical care were de-identified and entered into a unified case reporting form, detailing selected demographics, clinical findings and laboratory results (oligoclonal bands in cerebrospinal fluid and EBV antibody profile), first and subsequent attack characteristics, neuropsychological assessment outcomes and treatment information.

Annualised relapse rates (ARRs) on retrospective data were calculated as number of relapses per year, only including patients with minimum 6 months follow up after treatment initiation. If time to treatment was less than 6 months, pre-treatment ARR was calculated over a 6 month period. Neurological disability is described using the Expanded Disability Status Score (EDSS), a measure widely used to quantify adult MS disease progression [18], and if unavailable was retrospectively scored to estimate level of functioning two years from diagnosis, excluding measurements within 3 months of relapses.

Cognitive domains assessed were global intellectual functioning, academic attainment and visuospatial ability, using the Wechsler Intelligence Scale for Children, the Wechsler Individual Achievement Test, the Children’s Memory Scale, and the Beery–Buktenica Developmental Test of Visual–Motor Integration, respectively, assessed as part of standard care across both sites. Quality of life was assessed using PedsQL self-rating scores from routine psychology assessments; a validated measure of health-related quality of life in children with chronic health conditions [19].

### 2.3. Statistical Analysis

Mann–Whitney U tests were used to compare continuous variables between cohorts and paired t-tests for pre- and post-treatment relapse data. Univariate analysis of variance was used to control for time to neurocognitive assessment. All tests performed were two tailed; *p*-values lower than 0.05 were considered statistically significant and care was given to limiting multiple comparisons. A Bonferroni correction was used to correct for multiple comparisons when analysing neurocognitive data. Analyses were performed using SPSS Statistics, version 25.

### 2.4. Ethical Approval

All data collected utilised dataset from a collaborative study previously approved by GOSH Research and Development Department (reference 16NC10).

## 3. Results

### 3.1. Cohort Characteristics

A total of 24 patients were identified in the earlier cohort and 27 in the later. A total of 14 other patients initially diagnosed with MS subsequently tested positive for MOG antibodies, and hence were excluded from this study. There were no significant differences between cohorts in demographics or paraclinical markers, as shown in Table 1. In both cohorts, many patients had symptoms in more than one domain, with significantly fewer later-cohort patients experiencing isolated motor and cranial nerve involvement at onset. Time from first symptoms to diagnosis was similar between cohorts (median earlier cohort 10 months; median later 9 months, *p* = 0.94, Mann–Whitney U test; Figure 1A). Pre-treatment ARR was significantly higher in the earlier cohort.

### 3.2. Changes in Management

A total of 19/24 patients in the earlier cohort were commenced on DMTs compared to 24/27 in the later cohort (*p* = 0.29, Fisher’s Exact). Median time from diagnosis to initiation of first DMT was 9 months in the earlier cohort vs. 3.5 months in the later cohort (*p* = 0.013, Mann–Whitney U test; Figure 1B). Interferon beta-1a was the most commonly used initial DMT in both cohorts (79% earlier; 59% later). Figure 2 illustrates the increasing variety of DMTs used, with four new DMTs and two clinical trials introduced since 2010.

### 3.3. Disease Course

Out of the 43 patients ever started on DMT, 27 (63%) relapsed on treatment over a median follow-up period of 36 months. This encompasses 16/19 (84%) earlier-cohort patients on DMT relapsing vs. 11/24 (46%) in the later cohort. Mean ARR prior to DMT initiation was higher in the earlier cohort (mean ARR earlier 2.7; mean ARR later 1.7, Mann–Whitney U; *p* = 0.010). Mean ARR for entire treatment course after DMT initiation (mean duration 5.2 years earlier; 2.4 years later) was 0.74 in the early cohort and 0.37 in the later (Mann–Whitney U; *p* = 0.029). Improvement in ARR post-treatment was observed in both cohorts, with a reduction of 2.0 (2.7 to 0.74) in the earlier cohort (*p* = 0.001, Figure 1C), and 1.3 (1.7 to 0.37) in the later (*p* = 0.001, Figure 1D). Median EDSS two years from diagnosis was similar between cohorts (1.0 earlier, IQR 0.0–2.3; 1.0 later, IQR 0.0–1.5, Mann–Whitney U; *p* = 0.47).

### 3.4. Cognitive and Quality of Life Outcomes

In the earlier cohort, 58% of patients underwent neuropsychological assessment (n = 14), compared to 89% in the later (n = 24). For the 10 patients assessed multiple times, the assessment closest to 2 years from disease onset was used. Median time from disease onset to cognitive assessment was 9.5 months longer in the earlier cohort (28.5 months earlier; 19.0 months later, Mann–Whitney U; *p* = 0.049).

Across several neurocognitive tests analysed, the later cohort performed better than the earlier cohort with significantly higher Spelling (median −0.67. vs. 0.00, *p* = 0.041), Word Reading (median −0.25 vs. 0.60, *p* = 0.025) and Delayed Memory z-scores (median −1.00 vs. 0.00, *p* = 0.042). When adjusting for increased time to assessment in the earlier cohort, only Word Reading z-scores were significantly improved (*p* = 0.013) and there was no significant difference in Spelling (*p* = 0.099) or Delayed Memory (*p* = 0.13). Table 2 displays results of all neurocognitive tests analysed. A detailed map of each patient’s results is included in Data Table A1, demonstrating that of 11 patients with any cognitive impairment, 7 had impaired scores on at least 2 measures, with 4 of these impaired on at least 3 measures.

PedsQL scores were conducted on 2/24 patients in the earlier cohort and 13/27 in the later. Median PedsQL overall was 81.7 (IQR 65.3–92.4), corresponding approximately to normal scores observed in healthy populations [20].

## 4. Discussion

This retrospective analysis of POMS management over eight years demonstrates changes in both treatment approach and outcomes, with the later cohort receiving earlier treatment with a wider variety of DMTs. The earlier and later cohorts were comparable, with patients studied reflecting what we know of POMS, with most patients post-pubertal and an approximately 2:1 F:M ratio [4]. Differences in patients’ presenting symptoms between cohorts were observed, with fewer later-cohort patients presenting with isolated motor or cranial nerve involvement; the reasons for this are unclear but may partly result from differences in reporting and documentation of initial presentations.

### 4.1. Evolution of Care

In recent years, there has been a significant paradigm change in paediatric MS management. Over 14 DMTs are now available, many of which are highly efficacious, and almost all of which focus on the inflammatory component of MS [21]. This is particularly important when considering the more inflammatory profile of paediatric-onset disease compared to adult disease. The use of these DMTs in paediatric-onset MS has increased, reflected here in the later cohort [22]. The advent of paediatric clinical trials, with four patients in the later cohort enrolled, has also increased access to new treatments. Recently, the landmark completion of the PARADIGMS trial resulted in the paediatric licensing of Fingolimod in the US and Europe [23].

Additionally, largely informed by adult studies, treatment responses are now titrated towards multimodal evaluation of disease progression beyond utilising only clinical relapses. Evidence of disease activity is now measured clinically, radiologically and by absence of progression of disability, with a therapeutic aim of “no evidence of disease activity” [24,25]. The shorter time from diagnosis to DMT initiation in the later cohort may reflect the awareness of the benefits of early more effective intervention, with increased rapidity of treatment possibly contributing to reduced disease progression and optimised outcomes. This is also likely reflected in the lower pre-treatment ARR in the later cohort, which may be secondary to more rapid initiation of DMTs than in the earlier cohort.

Crucially, an improved management pathway relies on timely diagnosis. Revisions of the McDonald diagnostic criteria in 2010 [26] and 2017 [27] have facilitated this by allowing for confirmation of MS from a single scan of dissemination in time and space, with applicability in a paediatric population [28,29]. Nevertheless, in our cohort despite the use of the 2010 criteria in later patients, time to diagnosis remains as long as in the earlier. This may partly be addressed by the 2017 criteria and may also result from the important issue of continuing delays in referrals. To address this, NHS England has recently commissioned five specialist centres in England to provide care for children with MS [30], a model adapted from European and American centres of excellence. Direct referral to these centres serves as a means to facilitate early referral to address delays in diagnosis and enable timely access to high-quality care.

### 4.2. Changes in Outcomes

A variety of improved management including outcomes in the later cohort were observed, as illustrated in Figure 3. Importantly, as patients in the two cohorts had significantly different baseline relapse rates with a significantly lower pre-treatment ARR in the later cohort, reliable conclusions cannot be made about the effectiveness of DMTs being the determinant of improved outcomes observed. Nevertheless, relapse rates were significantly lower in both cohorts following DMT initiation when compared to baseline rates. Although the majority of patients relapsed on treatment, this may reflect greater tolerance of relapses during the earlier management epoch with inadequate escalation of DMT. In contrast, the relapse rate on treatment in the later cohort (46%) is comparable to a recent report in the literature of children started on first line DMTs, 43% of whom relapsed on treatment. Comparatively, a 19% relapse rate was reported for children treated with newer, higher efficacy agents demonstrating the importance of timely escalation of treatment to second line DMTs in order to further minimise relapses [31].

Consistent with what is known of POMS [4], both cohorts displayed low levels of physical disability. Median EDSS scores in both cohorts were equal to 1 (equivalent to no disability, minimal signs in one functional system). As DMTs have been shown to have a much greater impact on reducing relapse rates in POMS than reducing short to medium term disability progression [32], a longer follow up of up to 10 years is usually required to demonstrate this, as has been shown in adult cohorts [13]. Additionally, we could not comprehensively compare radiological responses to treatment across both cohorts as clinical relapses (instead of imaging) were the stronger determinant of disease monitoring in the earlier cohort. Treatment responses on MRI activity may be even more striking than the clinical responses observed [24,33], highlighting the value of combined measurement evaluating treatment efficacy.

Our neurocognitive findings demonstrated differences in a variety of areas known to be particularly impacted in POMS [34]. Most median scores were in the average range with many demonstrating normal cognitive abilities, yet a few patients performed poorly across several tests, particularly in the earlier cohort. Importantly, differences identified between cohorts may largely reflect the longer time from MS onset to testing in earlier patients. Nevertheless, when controlling for this, there was still significant improvement in Word Reading scores in the later cohort, suggesting potential benefits of earlier intervention with more effective treatment. Ultimately, without longitudinal data, cognition decline over time can only be extrapolated, with previous cohorts finding that most POMS patients experienced worsening cognitive outcomes over 5 years, with adverse consequences on academic and social achievements [35].

As a wider group, children with MS are known to report consistently lower quality of life than healthy controls [11]. However, this was not observed here. As so few patients in the earlier cohort had PedsQL scores, it was not possible to compare cohorts. However, we would expect that the changes observed in management and outcomes with more holistic care may have had an appreciable impact on quality of life. This holistic approach is also reflected in the increased numbers assessed using the PedsQL over time, with increased involvement of clinical psychology as part of routine care.

### 4.3. Limitations

The key limitation of this study was a consequence of the retrospective nature of the study design which precluded collection of data and outcome parameters across both cohorts to account for changes in the management of a complex disorder. Earlier management and diagnosis, more DMTs available and more holistic care may have all contributed to the improvements seen, alongside other unmeasured factors. As decision-making surrounding DMT is complex, it remains unclear as to what extent service factors are responsible for differing patterns of treatment between cohorts.

Like many studies on POMS, small sample sizes limit statistical power. In order to maximise sample size, cohorts were defined as all patients seen during each respective study period, as more precisely selected cohorts over a fixed time-point would have been too small to draw conclusions from. This is important when interpreting results, as cohorts have been selected to reflect service caseloads during study periods, rather than as controlled groups. Furthermore, a direct comparison of treatment efficacy cannot be drawn as patients in the cohorts had different baseline relapse rates. Improved robustness of data is needed for future research, with systemic data collection across all sites essential for overcoming many of the limitations encountered here.

## 5. Conclusions

This study provides support for the benefits of current service models for POMS, which involve early referral and multidisciplinary input resulting in prompt diagnosis and optimal escalation of treatment. This is in the context of an increasingly holistic approach to care focused on children’s wellbeing. Although we cannot ascribe improvements seen to any single factor, the cohort of patients treated earlier, more holistically and with newer DMTs are likely to experience multiple downstream effects predominantly in relapse rates. Secondary benefits would include fewer hospital admissions, fewer courses of steroid treatment, fewer missed school days and less residual disability resulting from irreversible neuronal loss [31].

There remain many challenges facing managing children with MS. Currently, there are two further RCTs of treatment in progress [36,37], with many more planned, particularly of the more highly effective agents. There are also ongoing open-label trials such as LemKids [38], which provide important data on the safety of DMTs. However, these are not addressing the key outcome that matters: is treatment preventing disability? Real-world data are crucial in addressing this issue and validating findings of clinical trials, as well as the inclusion of adolescents in adult trials in order to increase the scope of current research, as has been recently recommended [39]. Standardisation of outcomes is another important part of enabling valid comparisons of treatment approaches, with the target of “no evidence of disease activity” giving valuable clarity on the need to monitor relapses, disability progression and subclinical lesions on MRI to ensure that treatments are escalated appropriately. With follow up over several years necessary to adequately assess treatment efficacy, the continuity of data once patients transition into adult services is needed. Unified databases to integrate paediatric and adult records are an integral part of harmonising this data, with benefits both for individual patient care and in creating essential resources to be used in research.

## Figures and Tables

**Figure 1 children-07-00222-f001:**
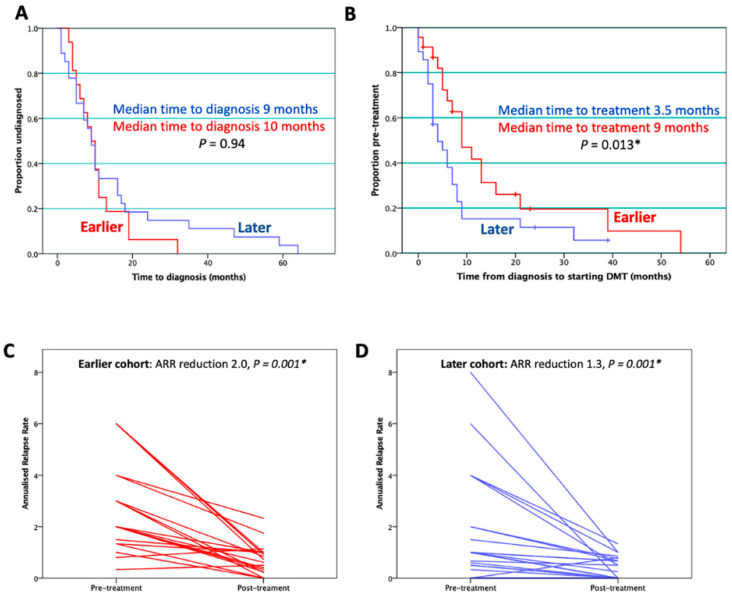
Effective treatment of POMS patients. (**A**) Time to diagnosis did not significantly differ between cohorts. (**B**) Time from diagnosis to first disease-modifying treatment was significantly shorter in the later cohort. Markers show duration of follow up for patients never started on DMT. (**C**,**D**) Annualised relapse rates (ARRs) decreased significantly in both cohorts after treatment initiation.

**Figure 2 children-07-00222-f002:**
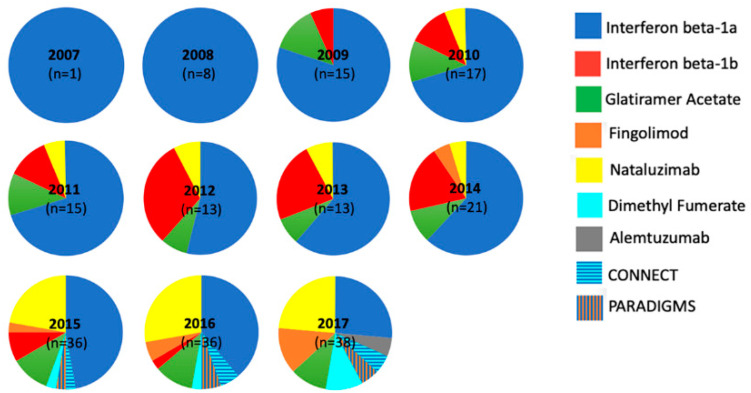
Disease-modifying treatments used in each year over the study period, demonstrating increasing variety over time. CONNECT (dimethyl fumarate vs. interferon beta-1a) and PARADIGMS (fingolimod vs. interferon beta-1a) denote clinical trials.

**Figure 3 children-07-00222-f003:**
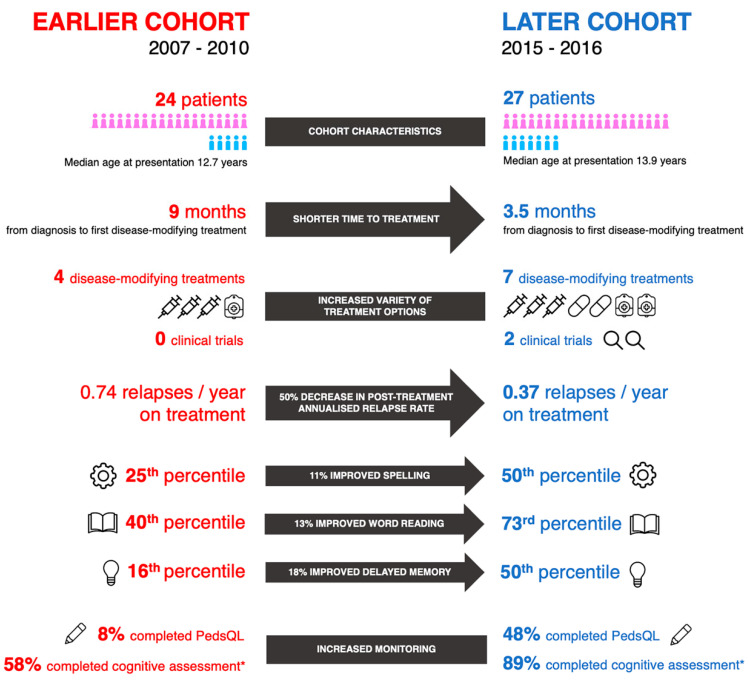
Key improvements over the study period when comparing the two cohorts. Arrows show direct compared outcomes, blocks demonstrate unquantified differences.

**Table 1 children-07-00222-t001:** Cohort characteristics, symptoms at onset and outcomes.

	2007–2010 (n = 24)	2015–2016 (n = 27)	*p*-Value
Age at presentation, median (IQR), y	12.7 (10.9–14.2)	13.9 (10.4–14.3)	0.56
Sex, M:F	1:3.8	1:2.9	0.75
Symptoms at onset (%)			
Vision	10/24 (42)	9/27 (33)	0.58
Isolated motor	12/24 (50)	3/27 (11)	0.005 *
Cerebellar syndrome	5/24 (21)	5/27 (19)	1.0
Sensory	10/24 (42)	9/27 (33)	0.58
Cranial nerve involvement	4/24 (17)	12/27 (44)	0.040 *
Paraclinical features (%)			
OCB	20/21 (95)	25/27 (93)	1.0
EBV IgG	8/8 (100)	23/23 (100)	1.0
Time to diagnosis, median (IQR), m	10 (5–13)	9 (5–17)	0.94
Follow-up time median (IQR), y	5.0 (2.5–7.5)	3.0 (2.0–5.0)	0.13
Patients not started on DMT (%)	5/24 (21)	3/27 (11)	0.29
Outcome			
Pre-treatment ARR, mean (IQR), *no. patients*	2.7 (1.3–4.0), *19/24*	1.7 (0.4–2.0), *24/27*	0.010 *
Post-treatment ARR, mean (IQR), *no. patients*	0.7 (0.3–1.0), *19/24*	0.4 (0.0–0.7), *24/27*	0.029 *
EDSS at 2 y, median (IQR), *no. patients*	1.0 (0.0–2.25), *18/24*	1.0 (0.0–1.5), *27/27*	0.47

Abbreviations: ARR, annualised relapse rate; DMT, disease-modifying treatment; EBV, Epstein–Barr virus; EDSS, Expanded Disability Status Scale; IQR, interquartile range; OCBs, oligoclonal bands. Data are presented as number (percentage) of patients unless otherwise indicated. Asterisks (*) indicate statistically significant improvements between cohorts.

**Table 2 children-07-00222-t002:** Results for all neurocognitive tests analysed.

NEUROCOGNITIVE TEST	Numbers Tested in 2007–2010	Numbers Tested in 2015–2016	Median z- Score in 2007–2010	Median z-Score in 2015–2016	Increase in Median z-Score
**Full-Scale IQ**	11	24	−1.50 **	−0.20	1.30
**Working Memory Index**	13	24	−0.90 *	0.00	0.90
**Processing Speed Index**	13	24	−1.50 **	−0.52	0.98
**Word Reading**	9	24	−0.25	0.60	0.85 #
**Pseudoword Decoding**	7	13	−0.52	0.66	1.18
**Spelling**	9	23	−0.67	0.00	0.67 #
**Numerical Operations**	10	22	−0.33	0.60	0.93
**Visual–Motor Integration**	7	24	−0.67	−0.33	0.34
**Visual Perception**	7	23	−0.25	−0.12	0.13
**Motor Coordination**	7	23	−2.00 **	−0.90 *	1.10
**Memory Stories Immediate**	8	19	−1.00 *	0.00	1.00
**Memory Stories Delayed**	8	19	−1.00 *	0.00	1.00 #

Single asterisks (*) denote low average (9–23rd percentile) median standard scores and double asterisks (**) denote borderline impaired standard scores (2–8th percentile). Increases in median z-scores marked with a hash (#) indicate statistically significant differences between cohorts. When adjusting for time from disease onset to assessment, only Word Reading scores were significantly different (*p* = 0.013).

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
