# Peer review of "Progress in the Management of Paediatric-Onset Multiple Sclerosis"

_children, 2020, doi:10.3390/children7110222_

Round 1
Reviewer 1 Report
This is a retrospective review of POMS patients managed at two paediatric neuroimmunology centres aiming to evaluate how recent advances have affected patient management and outcome. The authors compared two cohorts, seen 2007-2010 and 2015-2016, respectively, and describe demographic, clinical and neurocognitive data. Within this relatively short time span, they are able to demonstrate significant shifts in treatment practise, neuropsychological evaluation and outcome. Thus, results are both solid and of great interest. Being a retrospective study, some variables will be unknown but this is addressed adequately and handling of data is solid.
It is a very interesting manuscript which is well written and easy to follow, figure 3 is excellent. I have only a few remarks:
- In the results section, the authors claim that “There were no significant differences between cohorts in demographics, clinical or paraclinical markers, as shown in Table 1”. However, there are differences in symptom presentation and also in pre-treatment ARR. This needs to be altered in the text and addressed in the discussion concerning potential differences in cohorts and possible reasons for this.
- The authors also found a 63% relapse rate over a 36-month period, a high number when on adequate DMD treatment. It would be interesting to the reader to have this addressed in the discussion.
- Statistical analyses do not account for multiple analyses (15 in Table 1). Results do not appear to suffer from this, but this number is in general too high for not performing post-hoc analysis.
Author Response
Response to Reviewers’ Comments
We are grateful for the thorough comments received and believe that in addressing them, we have added to the quality of our manuscript. Our responses are outlined below:
Point 1: In the results section, the authors claim that “There were no significant differences between cohorts in demographics, clinical or paraclinical markers, as shown in Table 1”. However, there are differences in symptom presentation and also in pre-treatment ARR. This needs to be altered in the text and addressed in the discussion concerning potential differences in cohorts and possible reasons for this.
Response 1: We apologise for this error. We have altered the text in the results section (page 3, lines 104-108) to add “In both cohorts, many patients had symptoms in more than one domain with significantly fewer later cohort patients experiencing isolated motor and cranial nerve involvement at onset… Pre-treatment ARR was significantly higher in the earlier cohort.” The reason for the pre-treatment ARR has now been addressed (page 7, lines 196-197) and is likely to reflect management practices of slower initiation of DMTs, whilst the differences in presenting features are less clear and this we now raise in the discussion (see page 6, lines 178-180).
Point 2: The authors also found a 63% relapse rate over a 36-month period, a high number when on adequate DMD treatment. It would be interesting to the reader to have this addressed in the discussion.
Response 2: We thank the reviewer for raising this important observation which we now highlight and appropriately discuss the implication. Firstly, we add additional information that earlier cohort patients were experiencing more relapses on treatment compared to later cohort (84% vs 46%; page 5, lines 137-138).Secondly, we now explain in the discussion that the higher relapse rate whist on treatment is reflective of the tolerance of relapses during the earlier management epoch resulting in patients not being adequately escalated. By contrast, the relapses seen in the later epoch (46%) are comparable to a recent report in the literature of children who have been started on conventional first line DMTs but escalated significantly quicker (43%). We also now discuss the potential to improve this with newer more effective agents (page 7, lines 214-220).
Point 3: Statistical analyses do not account for multiple analyses (15 in Table 1). Results do not appear to suffer from this, but this number is in general too high for not performing post-hoc analysis.
Response 3: Thank you for highlighting this. We are indeed mindful that in most studies of this nature where a clear analysis plan is not determined, analysis can be susceptible to numerous post-hoc (and ad-hoc) analysis. As the reviewer also points out, parameters analysed were within limits; and we now state this pragmatic efforts in the methods section (see page 3, lines 93-94). A Bonferroni correction was used to correct for multiple comparisons when analysing neurocognitive data, which we overlooked describing in the methodology, which we have now rectified (see page 3, lines 94-95).
Reviewer 2 Report
The paper is short, sharp and on an area of great interest within the field of Paediatric neuroscience. It is a well written paper describing a retrospective analysis of two cohorts of children with POMS separated by a considerable time interval. As the authors report: the latter group had different treatments and in particular quicker access to disease modification therapy.
As the authors indicate, they are unable to make many conclusions from their study due to the lack of control within the different groups. In particular, although they described the earlier and multiple DMT, in the later group, the follow-up is not over a long enough timescale to detect significant changes in disability, and with many different therapies studied, it was not possible to rank or indicate the most appropriate stages in escalation. They also noted the difference in the first and second groups as to parameters prior to initiating DMT, making in difficult to draw conclusions on efficacy.
In terms of neuropsychological assessment they reported improvements in the latter cohort in one area only (having discounted 2 others) - word reading, but I am unsure as to whether they have undertaken a Bonferroni correction and if not, whether that might nullify the significance of the difference.
Although the authors described a large cohort of children originally and compare these to the subsequent one undertaking DMT earlier, and describe more rapid intervention / reducing relapse rate, they have not been able to show that these changes in treatment were beneficial and a consequence of earlier DMT.
Author Response
Response to Reviewers’ Comments
We are grateful for the thorough comments received and believe that in addressing them, we have added to the quality of our manuscript. Our responses are outlined below:
Point 1: As the authors indicate, they are unable to make many conclusions from their study due to the lack of control within the different groups. In particular, although they described the earlier and multiple DMT, in the later group, the follow-up is not over a long enough timescale to detect significant changes in disability, and with many different therapies studied, it was not possible to rank or indicate the most appropriate stages in escalation. They also noted the difference in the first and second groups as to parameters prior to initiating DMT, making in difficult to draw conclusions on efficacy.
Response 1: We certainly agree with the reviewer with regards to the key evaluation of the message of the paper.
Point 2: In terms of neuropsychological assessment they reported improvements in the latter cohort in one area only (having discounted 2 others) - word reading, but I am unsure as to whether they have undertaken a Bonferroni correction and if not, whether that might nullify the significance of the difference.
Response 2: We acknowledge this oversight also raised by reviewer 1, we have now clarified that a Bonferroni correction was undertaken (see response to point 3 for reviewer 1, and page 3, lines 93-95).
Point 3: Although the authors described a large cohort of children originally and compare these to the subsequent one undertaking DMT earlier, and describe more rapid intervention / reducing relapse rate, they have not been able to show that these changes in treatment were beneficial and a consequence of earlier DMT.
Response 3: Alas, it is as the reviewer has pointed out. We have discussed the adult experience here (page 8, lines 227-230, and page 9, lines 252-265) where longer-term follow-up of up to 10 years and significantly larger cohorts (544 patients in the He et al paper cited as reference 13) were required to detect a difference between the groups.
Round 2
Reviewer 2 Report
Thank you for clarifying the points made in the previous review. In addition, thank you for doing this in a timely manner and highlighting the amendments,
However, regarding point three, I note the authors have replied to my comments, and apologise if I was not clear enough. I would suggest that with such an eminent group of authors, with immense experience within the field of POMS, and this article being more educational than carrying high quality data on the outcome of randomised controlled trials, they should expand on their conclusions.
In particular, what would overcome the weaknesses of this paper? i.e. how to design trials that will tell the reader, rather than likely but not proven corellations, the outcome of differing DMT's at different points – ie efficacy and safety, the challenges and solutions to the very long-term process of data collection, in children that are going to transition through to multiple different services, mimicking the adult data, as well as the difficult problem of randomised controlled trials, when there are so many DMT's.
I note the challenge was similar for the haematological oncologists within the management of leukaemia, with dramatically improved results in the longer term established on a clear evidence base. i.e. rigorous database data collection, stratified randomised controlled trials with national coordination.
Author Response
Point 1: However, regarding point three, I note the authors have replied to my comments, and apologise if I was not clear enough. I would suggest that with such an eminent group of authors, with immense experience within the field of POMS, and this article being more educational than carrying high quality data on the outcome of randomised controlled trials, they should expand on their conclusions.
In particular, what would overcome the weaknesses of this paper? i.e. how to design trials that will tell the reader, rather than likely but not proven corellations, the outcome of differing DMT's at different points – ie efficacy and safety, the challenges and solutions to the very long-term process of data collection, in children that are going to transition through to multiple different services, mimicking the adult data, as well as the difficult problem of randomised controlled trials, when there are so many DMT's.
I note the challenge was similar for the haematological oncologists within the management of leukaemia, with dramatically improved results in the longer term established on a clear evidence base. i.e. rigorous database data collection, stratified randomised controlled trials with national coordination.
Response 1: Thank you for your further comments. We now fully understand the nature of the original comment as a suggestion to add more to the discussion about all the limitations of this study but also paediatric MS clinical trials as a whole. We now highlight a key document published by the International Paediatric MS Study Group (IPMSSG, reference 40) addressing this important question and have added an additional paragraph (page 9, lines 272-285) to elaborate on this point.